# Aligning the Goals Hybrid Model for the Diagnosis of Mental Health Quality

Wagner Silva Costa [1], Plácido R. Pinheiro [2,*], Nádia M. dos Santos [1] and Lucídio dos A. F. Cabral [3]

[1] Central Teresina Campus, Instituto Federal do Piauí, Teresina 64000-040, Brazil; wagnersc@ifpi.edu.br (W.S.C.); nadiaphb@gmail.com (N.M.d.S.)
[2] Graduate Program in Applied Informatics, University of Fortaleza (UNIFOR), Fortaleza 60811-905, Brazil
[3] Computer Center, Federal University of Paraíba, Paraíba 58058-600, Brazil; lucidiocabral@gmail.com
[*] Correspondence: placido@unifor.br

**Abstract:** The social distancing imposed by the COVID-19 pandemic has been described as the "greatest psychological experiment in the world". It has tested the human capacity to extract meaning from suffering and challenged individuals and society in Brazil and abroad to promote cohesion that cushions the impact of borderline experiences on mental life. In this context, a survey was conducted with teachers, administrative technicians, and outsourced employees at the Federal Institute of Piauí (IFPI). This educational institution offers professional and technological education in Piauí, Brazil. This study proposes a system for the early diagnosis of health quality during social distancing in the years 2020 and 2021, over the COVID-19 pandemic, combining multi-criteria decision support methodology, the Analytic Hierarchy Process (AHP) with machine learning algorithms (Random Forest, logistic regression, and Naïve Bayes). The hybrid approach of the machine learning algorithm with the AHP multi-criteria decision method with geometric mean accurately obtained a classification that stood out the most in the characteristics' performance concerning emotions and feelings. In 2020, the situation was reported as the SAME AS BEFORE, in which the hybrid AHP with Geographical Average with the machine learning Random Forest algorithm stands out, highlighting the atypical situation in the quality of life of the interviewees and the timely manner in which they realized that their mental health remained unchanged. After that, in 2021, the situation was reported as WORSE THAN BEFORE, in which the hybrid AHP with geometric mean with the machine learning Random Forest algorithm provided an absolute result.

**Keywords:** mental health; COVID-19; social distancing; AHP; Analytic Hierarchy Process; machine learning

## 1. Introduction

Some countries, intending to reduce the impacts of the COVID-19 pandemic, its peak incidence, and the number of deaths, have adopted measures to isolate suspected cases, closing schools and universities and social distancing from risk groups (elderly and others), as well as quarantining the entire population [1,2]. It is estimated that these measures tend to delay the spread of the pandemic, promote a lower incidence peak during a given period, and reduce the likelihood that the capacity of hospital beds, respirators, and other supplies will be inadequate in the face of a sudden increase in demand, which would be associated with a higher mortality rate among the general population [2].

People are sociable, and prolonged distance can cause significant psychological suffering in the community [3]. Social distancing measures are often considered unpleasant by those who experience them. The sudden change in daily activities (e.g., work routine, studies, and community life), sometimes without knowing when the return to "normal life" will occur, may cause suffering and insecurity since it is necessary to deal with the unpredictable future. In addition, the decrease in face-to-face interactions tends to generate

a feeling of social isolation, often accompanied by emotional isolation and the deprivation of freedom [4]. Commonly, patients in quarantine with confirmed or suspected COVID-19 cases may feel boredom, loneliness, and anger, among other mental disorders [5]. In situations of distancing and isolation, the population suffers from common malaise, such as impotence, boredom, loneliness, irritability, sadness, and various fears (e.g., of getting sick, dying, losing their means of subsistence, transmitting the virus). It all can lead to changes in appetite and sleep, family conflicts, and the excessive consumption of alcohol or illicit drugs. The elderly, especially those with cognitive decline or dementia, are identified as particularly vulnerable to emotional and behavioral changes [6,7].

The social distancing imposed by the COVID-19 pandemic has been described as the "greatest psychological experiment in the world". It tested the human capacity to extract meaning from suffering. It challenged individuals and society in Brazil and abroad to promote cohesion that cushions the impact of borderline experiences on mental life.

This work consists of a study conducted on teachers, administrative technicians, and outsourced employees at the Federal Institute of Piauí (IFPI). This educational institution offers professional and technological education in Piauí (Brazil).

The study proposes a system for the early diagnosis of health quality during social distancing in 2020 and 2021. It was carried out during the COVID-19 pandemic through hybrid algorithms combining machine learning algorithms (Random Forest, logistic regression, and Naïve Bayes) with the multi-criteria decision support methodology and the Analytic Hierarchy Process (AHP), improved by the geometric mean method.

In Section 2, we will discuss the definition of the problem itself and the methods used to generate the hybrid algorithms for identifying the state of mental health. Section 3 will present the data and results of applying hybrid algorithms in this study. In Section 4, we will evaluate the proposed models and make predictions with the trained data to be used as a research source in future pandemics. Finally, in Section 5, we will make the research conclusion and discuss the future works of this study.

## 2. Problem Definition and Optimization Model

This work conducted a study of teachers, administrative technicians, and outsourced employees at the Federal Institute of Piauí (IFPI). This multi-curriculum and multi-campus institution of higher, primary, and technological education offers vocational education and training (VET) in different teaching modalities in Piauí, Brazil. Besides de Rectory, it currently has 20 campuses, approximately 2230 teaching and administrative staff members, and 23,396 students.

In March 2020, given the confirmation of the coronavirus pandemic, the Federal Institute of Education, Science, and Technology of Piauí (IFPI), through Regulatory Ordinance n° 853 [8,9], suspended in-person classroom activities.

In June 2020, to systematize knowledge about the mental health implications and psychological interventions for the IFPI students and employees due to social distancing resulting from the COVID-19 pandemic, a team of psychologists from the campuses Angical, Floriano, Parnaíba, Paulistana, Pedro II, Picos, Piripiri, Reitoria, Teresina Central, and Teresina Zona Sul prepared a questionnaire about mental health. In this survey, 3255 questionnaires were sent to the institutional email of students and IFPI staff to be answered online on the Google Docs platform. The questionnaire was prepared without the need of either the IFPI Ethics Committee or the Research Committee involving Human Beings due to the urgency of obtaining information on the psychological disorders the agents involved in the study suffered.

In March and April 2021, more than a year after the beginning of remote work and remote teaching activities, there was a need to reassess and observe the extension of the pandemic and social distancing on aspects of the mental health of the school community. Therefore, the same research was conducted on the same campuses with a total of 2168 respondents in the second questionnaire application (2021).

The extension of the pandemic period and the uncertainties regarding the return of in-person classroom activities impacted the school community's physical and emotional health, which might have caused psychopathological manifestations. In addition to the psychological implications directly related to the disease, measures to contain the pandemic may include mental health risk factors.

Hybrid models are presented to propose a system for early diagnosis of health quality during social distancing in 2020 and 2021 over the COVID-19 pandemic. These models combine machine learning algorithms (Random Forest, logistic regression, and Naïve Bayes) with the methodology of multi-criteria decision support and the Analytic Hierarchy Process (AHP), improved with the geometric mean method.

From the authors' point of view, choosing a multi-criteria method among those available, applied to a given context, should suit the characteristics of the problem at hand. An important point will be evaluating the problem, the objects of the decision, and the available information. According to [10], the method selection should result from evaluating the chosen parameters, the data's type and precision, the decision-maker's way of thinking, and his/her knowledge of the problem. It is also important to note that the direct consequence of choosing different methods is that the results can be discordant and contradictory. According to [10], the evaluation should be simple since the observed differences are much more related to the diversity of results than to contradictions. In addition, some criteria allow for the validation of the chosen method. In the problem **"Multi-criteria Model for Diagnosis of Mental Health"**, the application of the AHP method with geometric mean is due to the acceptance of the method by the decision-maker. In other words, the questions presented to the decision-maker made sense to him/her, and he/she had confidence in answering them.

Furthermore, the need to evaluate the acceptance of the data, its properties used by the method, and whether the result supported the decision process was highlighted. Secondary issues, such as the existence of tools such as *Expert Choice* and accessible software formats, according to [11], were also observed, as they allowed for greater integration with the problem addressed. According to [11,12], the multi-criteria decision support methodology has several methods that can be applied to the most diverse problems. The principles and concepts of machine learning algorithms (Random Forest, logistic regression, and Naïve Bayes) will be shared with the multi-criteria decision method (AHP), which is used in this study to identify the quality of mental health during social distancing in the years 2020 and 2021 during the COVID-19 pandemic.

### 2.1. Machine Learning

This section focuses on the machine learning algorithms used in this study to build hybrid algorithms: Random Forest, logistic regression, and Naïve Bayes. The hybrid algorithms had their initial weights generated through a multi-criteria decision method (AHP) with the geometric mean to identify the quality of mental health during social distancing in the years 2020 and 2021 in the COVID-19 pandemic. In addition, the prediction capacity of the hybrid algorithms may be used in other pandemics regarding mental health.

Some studies have been carried out on mental health involving the COVID-19 pandemic and machine learning algorithms, such as the study by [13] and the machine learning-based predictive modeling of anxiety and depression symptoms during the COVID-19 pandemic by [14].

In machine learning, classification models aim to associate a category or type with a given observation to answer the following question: is this **"A" or "B"**? These are models developed from supervised learning, which is the construction of a model for the prediction or estimation of an **"output"** based on one or more **"inputs"** [15].

The independent variables (**inputs**) can be continuous, categorical, or both in classification models. The dependent variable (**output**) is categorical and usually dichotomous (**"A" or "B"**). Depending on the classification problem, an output that is not dichotomous can be converted to be so. Model development includes steps such as training and evaluation or

performance tests. During training, the goal is to obtain data that satisfactorily contemplate the problem's complexity so that an accurate predictive model can be developed from logistic regression, Naïve Bayes, and other classifiers based on machine learning. During the evaluation stage, the predictive capacity of the built model is finally tested using new data not associated with the previous training stage [15].

### 2.1.1. Logistic Regression

Logistic regression is a generalized linear model whose resulting classification estimates the probability of an event occurring due to a set of independent predictor variables, which can be qualitative, quantitative, or both. This operational mode results in a dichotomous classification of the occurrence or non-occurrence of an event. Its probability value lies between zero and one [16].

Logistic regression is a generalized linear model with three components. First, a random component is related to the dependent variable's probability distribution (response). Second is a systematic component that linearly relates the independent variables to the respective parameters. Third, a link function (**logit**) that relates the linear predictors of the model (systematic component) to the expected values of the response variable (random component). The determination of the parameters (regression coefficients) is performed using the maximum likelihood method, which generates values that maximize the likelihood function and generally present consistent mathematical properties [16].

As mentioned, the response variable (*y*) in logistic regression is dichotomous; two values are attributed to it: one for the event of interest, called a success, and zero for the complementary event, called failure. The probability of success is provided by $\pi_i$, and $1 - \pi$i provides the probability of failure. Considering a series of independent random variables, $x_1, x_2, x_3, \dots, x_n$, and a vector, $\beta = \beta_0, \beta_1, \beta_2, \dots, \beta_p$ formed by unknown model parameters, the probability of success is provided by

$$\pi_i = \frac{exp(\beta 0 + \beta 1 x i 1 + \beta 2 1 x i 2 + \dots + \beta p x i p)}{1 + exp(\beta 0 + \beta 1 x i 1 + \beta 2 1 x i 2 + \dots + \beta p x i p)} \tag{1}$$

The probability of failure is provided by

$$\mathbf{1} - \boldsymbol{\pi_i} = \frac{1}{1 + exp(\beta 0 + \beta 1 x i 1 + \beta 2 1 x i 2 + \dots + \beta p x i p)} \tag{2}$$

The *logit* for the logistic regression model is characterized by

$$g(x_1) = \ln\left[\frac{\pi_i}{1 - \pi_i}\right] = x^{T_i} \beta = \beta_0 + \sum_{j=1}^{p} \beta j x i j \tag{3}$$

And the log-likelihood function can be written as

$$z(\beta) = \sum_{i=1}^{n} [y i x T i \beta - \ln(1 + exp\{x T i \beta\})] \tag{4}$$

To evaluate the fit of the models obtained by logistic regression, one of the recommended methods is the use of the deviance function (*deviance*, ***D***). A model's deviance is defined as the deviation of this model regarding the saturated model, in which all parameters fit perfectly to all observations, according to the definition.

$$D = -2ln\frac{thelikelihoodofthefittedmodel}{thelikelihoodofthesaturatedmodel} \tag{5}$$

The numerator is the likelihood function of the adjusted model under discussion, and the denominator is the likelihood function of the saturated model. Considering that the simplest model is called the null model, formed only by the parameter $\beta_0$, the deviance is used to measure the discrepancy of an intermediate model of **p** parameters regarding the saturated model. The smaller the deviance, the better the fit of the model. Among

nested models, tests such as the likelihood ratio can assess the significance of a fit difference via deviance. This test is also used to verify the significance of the contribution of each predictor variable for the fit in a given model [16].

In regression models, predictive variables can be selected from iterative methods such as stepwise selection. The goal is to obtain parsimonious models that combine an optimal fit with the smallest possible number of variables [16]. During the iterations in the stepwise selection, one of the methods used to evaluate the statistical model is the *Akaike Information Criterion* (AIC). This criterion considers the model's fit and simplicity, penalizing models with more variables. Since AIC is a measure related to the loss of fit of a given model, the smaller this value is, the better the fit of the model [17].

$$AIC = -2log(L_p) + 2(p) \qquad (6)$$

Hierarchical binary logistic regression was used to address the impacts of COVID-19 on mental health (a sample from Rio Grande do Sul [18,19]) and also to identify the association between stress and aspects related to the COVID-19 pandemic using multiple logistic regression [9].

Next, this study will discuss the machine learning algorithm called Random Forest.

### 2.1.2. Random Forest

In machine learning, from the same training dataset, it is expected that the combination of results from several classifiers will improve the predictive performance and confidence in decision-making compared with the analysis of a single classifier. Therefore, there is an interest in researching and developing multiple predictive model methods (ensemble methods) characterized by generating many classifiers and combining their results. The Random Forest algorithm is an example of an ensemble method that uses decision tree classifiers [20].

According to the bagging method, the Random Forest produces many decision trees in the training phase. The mentioned method consists of the parallel creation and learning of predictors (each model is built independently) from the repeated generation of samples with replacements of the same size as the original dataset (bootstrap sampling). The bagging method in training aims to reduce the complexity of the models to avoid overfitting the data with very complex models. Furthermore, bagging reduces the variance that interferes with the performance of unstable generated predictors [21].

The Random Forest algorithm adds randomness to the model when creating trees, as it searches for the best characteristics to partition nodes based on random subsets of variables. This procedure generates diversity, which usually leads to the formation of better ensemble predictors. In the end, each classification tree is identified as a predictor component. Consequently, Random Forest builds its decision by counting the votes of the predictor components in each class and then selects the winning class in terms of the number of accumulated votes among all the "trees of the forest" [21].

The Random Forest algorithm has been used to approach death prediction and the importance of clinical features in older people with COVID-19 [22]. It has also been used for the COVID-19 rapid test result by combining the Web system with the Random Forest algorithm and the result of blood exams [23].

Next, this study will address the machine learning algorithm called Naïve Bayes.

### 2.1.3. Naïve Bayes

Naïve Bayes is a probabilistic classification method based on the theorem of Thomas Bayes, which determines the probability of an event, *B*, occurring on the condition that *A* has already occurred, as defined by

$$\mathbf{P(B|A)} = \frac{P(A/B)P(B)}{P(A)}$$

where **P(B|A)** is the conditional probability of **B** occurring given that **A** occurred; *P(B)* and *P(A)* are the probabilities of the occurrence of **B** and **A**, respectively; and **P(B|A)** is the conditional probability of **A** occurring given that **B** occurred [24].

The Naïve Bayes algorithm is called "Naïve" because it assumes that the predictor variables (qualitative, quantitative, or both) are conditionally independent. Therefore, the information about an event is not informative about any other. It is one of the most used statistical learning methods in classification problems due to its simplicity and predictive capacity [24]. A priori probabilities are those previously used in the NB algorithm and are associated with the occurrence frequencies of classes in the training dataset. The logic underlying a priori probabilities is that the occurrence frequencies of classes in the training data are similar to those in the test dataset. Therefore, the prediction of an outcome is influenced not only by the predictor variables but also by the prevalence of the outcome. Conditional probabilities are calculated for each variable. Thus, considering that Naïve Bayes uses Bayes' Theorem as a principle, we have [24]

$$\mathbf{P(y_i \,|\, x)} = \frac{P(x/yi)P(yi)}{P(x)} \tag{8}$$

where **P(y$_i$|x)** is the *posterior* probability, which is the probability of a given observation, with its respective predictor variables *(x)*, belonging to class *y$_i$*; **P(x|y$_i$)** is the conditional probability (likelihood), which is the probability of verifying observations that belong to a given class. It is decomposed into probabilities concerning each of the model's predictor variables, which are multiplied among themselves, and in this case, P (x$_1$ | y$_1$) x ... x P(x$_i$ |y$_i$); P(y$_i$) is the probability of the referred class (prevalence); and P(x) is the occurrence probability of the predictor variables under discussion. Ultimately, the denominator can be ignored since it is the same for all classes. Unlike logistic regression, in Naïve Bayes, there is no post-training variable selection phase [15,24].

Throughout this section, the principles and concepts of a multi-criteria decision method (AHP) (resulting from the geometric mean) and machine learning algorithms (Random Forest, logistic regression, and Naïve Bayes) will be presented. They are used in this study to identify the quality of mental health during social distancing in 2020 and 2021 caused by the COVID-19 pandemic. At the end of this section, the action strategy included in this study will be presented.

The Naïve Bayes algorithm was used to address machine learning techniques to predict depression and anxiety in pregnant and postpartum women during the COVID-19 pandemic, including a cross-sectional regional study [25].

The AHP (Analytic Hierarchy Process) method is described below.

### 2.2. Analytic Hierarchy Process (AHP)

Before explaining the multi-criteria method (the Analytic Hierarchy Process—AHP), we will expound on the scientific basis of the origin of this method.

The word decision comes from the Latin *decidere*, a combination of two words: *de* (meaning *off*) and *caedere* (meaning *cut*). Consequently, the word "decision" can be defined as a process in which one wants to stop interruptions to let it flow [26].

When faced with the need for a decision, the responsible individual, better known as the decision-maker, stops and only continues the process when he effectively decides. For that reason, a taken decision allows the process to return to its flow based on the new decision [26].

Historically, human beings have always pursued support for decision-making. In the early days of human civilization, people tried to obtain the support of entities, known as deities, in decisions that resulted in success. To establish contact with said entities, human beings used individuals who had the "gift of dialoguing" with said deities by offering sacrifices that appeased the deities' anger and sensitized them to the claims of their worshipers and plaintiffs [26].

According to Bana and Costa et al. [14], the main stages of the decision-support process are the following:

(a) STRUCTURING: This deals with the formulation of the problem and the identification of objectives. This phase aims to identify, characterize, and organize the relevant factors in the decision-support process.

(b) EVALUATION: This allows for the subdivision of a subphase partial evaluation of actions (alternatives) according to each point of view (criteria) and an overall evaluation subphase considering the various partial evaluations.

(c) RECOMMENDATION: In this phase, sensitivity and robustness analyses are carried out to verify whether changes in the parameters of the evaluation model interfere with the final result. It is a fundamental phase that contributes to generating knowledge about the problem, increasing the confidence of the decision-maker in the obtained results.

Decision analysis is intended to improve the quality of decisions and communication between physicians, patients, and other healthcare professionals [27,28]. It is determined to work with choices under uncertainty; therefore, it is naturally adapted to the clinical setting. Unlike most everyday decisions, many healthcare decisions have substantial consequences and involve significant doubts. Uncertainties may be related to the diagnosis, the accuracy of diagnostic scripts, the background of the disease, the effects of a patient's treatment, or the effects of an intervention on a group or population.

That said, decision-making in seeking a diagnosis for a health disorder should not be treated informally. Nevertheless, this practice is quite common nowadays, most likely due to the short time spent in diagnostic search processes, notably in the public health service. These actions can compromise the diagnosis quality; the health of sick individuals; and, consequently, the health system, generating a waste of resources and increased expenses.

On the other hand, the present study aims, among other aspects, to show the positive impact of using information technology to diagnose disorders in mental health. Undoubtedly, information technologies have played a fundamental role in establishing diagnoses in the health field. Additionally, articles describe the various approaches that try to present the optimal way to use information technology in healthcare, mainly through support systems for diagnosing mental illness [28].

In order to establish a diagnosis in psychology and psychiatry, studies, experiments, and careful observations of the symptoms that lead to the cause of the evidenced disorder are necessary. One of the most significant difficulties in achieving a correct diagnosis is the complexity of the evidenced factors. In addition to a large amount of information, the specialist must consider cultural, biological, and psychosocial matters; quality of information; and signs and symptoms common to several illnesses. All these factors complicate the decision-making process, making it challenging to model, as it also considers unstructured information.

The methodology to support multi-criteria decision-making has a lot to add to the diagnostic processes of psychology and psychiatry since it provides the decision-maker with techniques and tools to structure the controlled events under analysis. It also allows for the hierarchization of these events for proper classification in order of the degree of importance of each one in the decision-making process of seeking the diagnosis.

In the next section, we will deal with the multi-criteria AHP method.

### 2.2.1. Multi-Criteria Methodology

Before the first half of the 20th century, mathematics was heavily used to make decisions under random conditions, which unacceptably contributed to risk situations. With the end of the Second World War and the Allied troops' experience concerning solutions to military logistical problems, many research institutions dedicated themselves to analyzing and preparing decisions using Operations Research [28,29].

The traditional problem-solving methods proposed by the Operations Research field try to fit problems into categories, and once they are classified, they can be solved through

standard procedures. Therefore, these methods focus on the choice of alternatives and the optimal solution to be found. Over time, it was observed that the urge to deal with even more complex situations required incorporating subjective aspects when making decisions. These aspects must still be explicit and quantified somehow. The multi-criteria decision support method aims to offer the decision-maker tools to solve problems with one or several alternatives [30–32].

Among the advantages of using the multi-criteria methodology are the following [32]:

- It is easy to use for non-specialists, preferably transformed into a computer program that is as user-friendly as possible, featuring visual graphic resources;
- It constitutes a logical and transparent method;
- It enables freedom from ambiguity for input data interpretations;
- It encompasses both quantitative and qualitative criteria;
- It values judgments;
- It allows the decision-maker to have algorithms that enable the use of criteria that are independent of each other, such as algorithms that help in problems in which the evaluation criteria are interdependent, and, similarly, it can deal with alternatives that are independent of each other;
- It incorporates human behavior issues into decision-making processes.

One of the main goals of multi-criteria decision analysis is to help decision-makers organize and synthesize information while feeling comfortable and confident when making a decision. Through human decisions, multi-criteria methods are put into operation.

Based on his knowledge of the problem, the decision-maker leads to the most appropriate course of action [32]. The following factors are essential to identifying the decision-maker's preference system:

- Considering the subjectivity of decision-makers, that is, the individual perceptions and envisioning involved in the aspects of problems, decision-makers find it most challenging to explain their perceptions;
- Structuring the problem according to the shared vision;
- Identifying common points of view;
- Knowing where decision-makers are inconsistent;
- Checking what can be changed and for what reason.

Decision-makers' preferences are crucial for structuring and modeling a multi-criteria decision problem. Participants in the decision-making process who deem it convenient to use the multi-criteria methodology to help structure their problems and, subsequently, prioritize or choose feasible alternatives must first [32]

- Define and structure the problem;
- Define the set of criteria or attributes or both, that will be used to rank the alternatives;
- Choose whether to use discrete or continuous methods; in cases of opting for discrete methods (conceived to work with a finite number of alternatives), it must favor the use of methods either from the French School or the American School;
- Identify the preference system of the decision-makers;
- Choose the aggregation procedure.

In the modeling phase of a problem, while using multi-criteria methods as a support, it is crucial to take into account [32]

- The choice of alternatives;
- The construction of criteria and information aggregation;
- The classification of the alternatives in which the dominance of the groups is identified;
- The ordering of a classification hierarchy among the alternatives.
- The structuring phase of a problem can be divided as follows [32]:
- The structure and composition of the components;
- The analysis;
- The synthesis of information.

The decision tree (Figure 1) is generic for all methods addressed in multi-criteria modeling. However, these models differ in some aspects, such as the problem detail level, the applied techniques, and the aggregation methods [32].

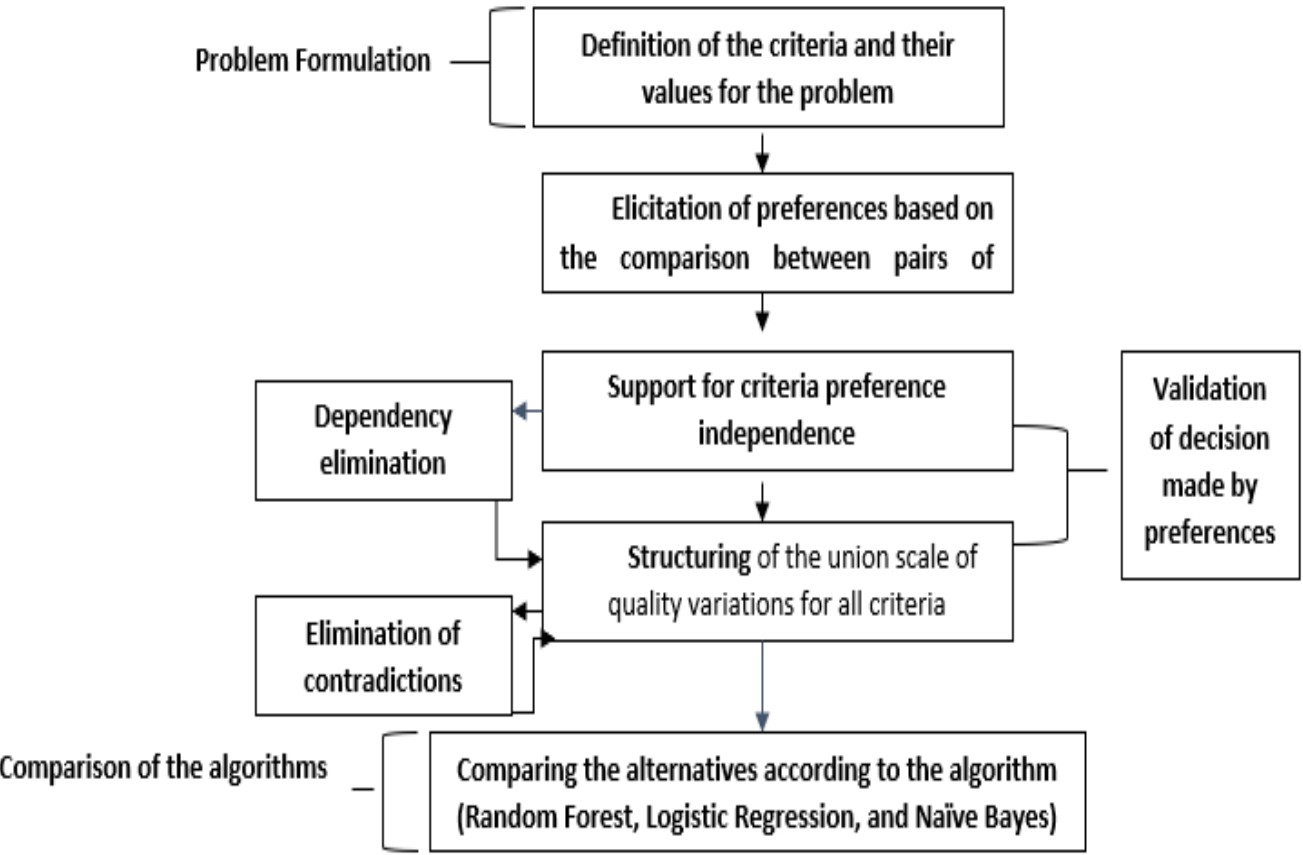

**Figure 1.** A theoretical framework to analyze the consistency of the outputs within the framework.

Multi-criteria analysis techniques emerged in the 1970s and 1980s, replacing the classic OR models that emerged in the 1950s, which pursued solutions to complex management problems [32].

Since its emergence, the study of decision problems embedded in a complex environment has been a subject of concern for researchers [33]. It highlights the existence of some methods applied to decision problems with multiple criteria within the scope of Operations Research, from which the field of study, multi-criteria decision support (MCDS), emerged.

MCDS methods are applied in various areas where one wants to select, order, classify, or describe existing alternatives in a decision-making process in the presence of multiple quantitative or qualitative criteria [34–41].

The multi-criteria decision support approach is characterized as a set of methods highlighting a problem. The alternatives are evaluated by multiple and conflicting criteria, helping people and organizations make decisions. The multi-criteria approach does not present an ideal solution to the problems but rather presents, among all possible decision alternatives, the most coherent one [38].

The multi-criteria decision support method has scientific and subjective aspects. It can aggregate all the necessary characteristics, including the non-quantitative ones. Additionally, it allows for transparency and the systematization of the process regarding decision-making problems [38]. The principle of multi-criteria support in the decision process is to establish a relationship in the preferences between the alternatives being evaluated under several criteria [37].

The study of decision problems [38] from the multi-criteria perspective does not aim to present the decision-maker with a specific solution to the problem but rather to support the decision process by recommending actions or courses of action to those who will make the decision [39].

Quirino highlights the emergence of two currents of thought that divide multi-criteria methods: multi-criteria decision-making (MCDM) and multi-criteria decision aid (MCDA). The first one emphasizes decision-making based on more analytical procedures in the search for a solution to the problem. In contrast, the second one emphasizes decision-making support based on a constructivist framework [40].

Dutra [41] discusses the two currents deeply, submitting the main differences according to Table 1.

**Table 1.** Differences between the MCDM and MCDA approaches.

| MCDM | MCDA |
| --- | --- |
| Existence of a well-defined set, A; the existence of a decision-maker, D | The border of A is diffuse and can be modified during the process. There is no decision-maker, D, but rather a set of actors participating in the decision-making process |
| Existence of a well-defined preference model in the mind of the decision-maker, D | Preferences are rarely well-defined, which are uncertainties, partial knowledge, conflict, and contradictions |
| Unambiguous data | It recognizes the data's ambiguity, often inaccurately or arbitrarily defined |
| Existence of an optimal solution to a well-defined mathematical problem | It is impossible to determine whether a solution is good or bad by considering only the mathematical model since cultural, pedagogical, and situational aspects affect the decision |

The two currents are represented in two schools of multi-criteria methods: the American School, which prefers MCDM, and the French one, also called the European School, which uses MCDA more [42].

According to utility theory, decision problems can be mathematically modeled by maximizing a function called the utility function, which is theoretically capable of representing the utility of each alternative for the decision-maker. Each alternative is assigned a grade (ordinal scalar value) through this function, which allows for ordering all alternatives from the best to the worst. The preferred alternative—the most useful one—is, therefore, the one with the highest score. The methods of the American School are characterized by helping the decision-maker build a utility function according to his/her preferences. It is based on an axiomatic theory that ensures the existence of this function.

The American School's main methods classify them according to the nature of their output data into three categories corresponding to the three types of decision problems: $P_1$, $P_2$, or $P_3$. The methods that generate an ordering of alternatives according to the decision-maker's preference are associated with $P_1$; the methods whose output data only indicate the favorite final solution to the problem are associated with $P_2$; and the methods that sort alternatives into predefined categories are associated with $P_3$. However, it is interesting to remember that the methods of category $P_1$ can also be used to determine a single final solution to the problem by selecting the best alternative in the ranking generated by it.

Hence, adopting the rationalist paradigm, the AHP fits into the MCDM or American School method. The American School's multi-criteria methods have, as a theoretical basis, the notion of aggregating all the information about the problem intended to be solved by employing a grand synthesis. Among the significant examples of this hierarchical analysis method, the most popular is the AHP method [42].

Therefore, the choice of method will depend on several factors, highlighting the characteristics of the analyzed problem, the considered context, the preference structure of the decision-maker, and the problem itself [39].

Problems involving multi-criteria have several active agents, so their definition is merely didactic [43]. Below are some concepts related to these acting agents in multi-criteria decisions:

**Decision-maker**: The agent has the power and responsibility to ratify the decision, assuming the consequences for this act, whether positive or negative. The decision-maker can be an individual or a group of people who establish the limits of the problem, specify the objectives to be achieved, and issue judgments. Not all decision-makers have decision-making power. Therefore, it is essential to distinguish the degree of influence of decision-makers in the decision-making process [44,45].

**Analyst**: This is the agent who interprets and quantifies the decision-makers' opinions, structures the problem, elaborates the mathematical model, and presents the results to the decision-maker. Constant dialog and interaction with decision-makers in a steady learning process are necessary [46].

**Model**: This is a simplified representation of reality through rules and mathematical operations that transform the decision-maker's preferences and opinions into a quantitative result [46].

**Alternative**: This is a potential action and constitutes the decision's object or the one directed to support it. It is identified at the beginning of the decision-making process or during it. It may become a solution to the problem under study [44].

**Criterion**: This is a function, $g$, defined in a set, $A$, that assigns ordering values from set $A$ and represents the preferences of the decision-maker from a certain point of view [47]. According to Morais [46], a problem with several criteria will be defined as $g_1, g_2, \ldots, g_j, \ldots, g_n$. The evaluation of an action, "a", according to the criterion, "$j$", is represented by $g_j(a)$. Representing different points of view (aspects, factors, characteristics) with the help of a criteria family, $F = \{g_1, \ldots, g_j, \ldots, g_n\}$, constitutes one of the most delicate parts of formulating decision problems. The criteria are classified according to the verified preference structure, as shown in Table 2.

**Table 2.** Criteria properties.

| Criterion | Preference Structure | Description |
|---|---|---|
| True criterion | Complete preorder (traditional model) | Any difference implies a strict preference |
| Quasi-criterion | Semi-ordered (threshold model) | There is a constant indecision zone between indifference and strict preference |
| Interval criterion | Interval order (variable threshold model) | There is a variable indecision zone between indifference and strict preference over the scale |
| Pseudo-criterion | Pseudo-order (double threshold model) | A sudden shift from indifference to strict preference is avoided, with a hesitation zone represented by weak preference |

**Criterion:** This is a function, $g$, defined in a set, $A$, which assigns ordering values from the set, $A$, and represents the preferences of the decision-maker from a certain point of view [47].

Concerning the dominance relationship, it occurs when two elements, "a" and "*b*", belong to set $A$, in the way that "a" dominates "*b*" (*aDb*) if, and only if, $g_1(a) \geq g_1(b)$, being that $j = 1, 2, \ldots n$, in which at least one of the inequalities is strictly preferred. It can be noted that the dominance ratio of "*a*" in "b" is characterized by being a strict partial order,

that is, an asymmetric and transitive relation. If **"a"** dominates **"b"**, **"a"** is superior to **"b"** in all problem criteria [47].

The essential concepts about the agents acting in multi-criteria decisions are

**Efficient Action**: This is an action (or alternative); *"a"* is considered efficient if, and only if, there is no other action from set *A* in which *a* dominates. The set of efficient actions of *A* can be *A* itself if the dominance relation is empty. It is generally considered a set that contains engaging actions to be analyzed in greater depth, even if there are no good reasons to disregard the inefficient ones [47].

**Decision Matrix**: Also called the *evaluation matrix*, in the decision matrix, each line expresses the measures of the evaluations of the alternative, *i*, concerning *n* considered criteria. Each column, in turn, expresses the evaluation measures of *m* alternatives concerning criterion *j*. Assuming that $a_{ij}$ represents the evaluation of the alternative (or action), $A_i$, it belongs to the set of potential actions: *A*, [$a_{ij}$]; according to criterion $g_j$, a matrix similar to Table 3 below can be constructed [46]:

**Table 3.** Decision matrix.

| Criteria ➡ | $g_1$ | $g_2$ | … … … …. | $g_J$ | … … … …. | $g_m$ |
|---|---|---|---|---|---|---|
| Limits ➡ | $q_1, p_1$ | $q_2, p_2$ | … … … …. | $q_j, p_j$ | … … … …. | $q_n, p_n$ |
| Alternatives ⬇ | | | | | | |
| $A_1$ | $a_{11}$ | $a_{12}$ | … … … …. | $a_{1j}$ | … … … …. | $a_{1n}$ |
| $A_2$ | $a_{21}$ | $a_{22}$ | … … … …. | $a_{2j}$ | … … … …. | $a_{2n}$ |
| … … … … | … … … …. | … … … …. | … … … …. | … … … …. | … … … …. | … … … …. |
| $A_i$ | $a_{i1}$ | $a_{i2}$ | … … … …. | $a_{ij}$ | … … … …. | $a_{in}$ |
| … … … .. | … … … …. | … … … …. | … … … …. | … … … …. | … … … …. | … … … …. |
| $A_m$ | $a_{m1}$ | $a_{m2}$ | … … … …. | $a_{mj}$ | … … … …. | $a_{mn}$ |
| $A_1$ | $a_{11}$ | $a_{12}$ | … … … …. | $a_{1j}$ | … … … …. | $a_{1n}$ |

### 2.2.2. The Classic Analytic Hierarchy Process (AHP) Method

Whichever model is defined to identify the quality of mental health in the situation of social distancing experienced during the COVID-19 pandemic in 2020 and 2021, the decision-maker will face a resolute state with a considerable volume of intersecting information and parallel and competing issues. The Analytic Hierarchy Process method (AHP), developed by Thomas L. Saaty in 1991, was chosen to analyze this complex scenario and to support the decision that will be applied in the machine learning algorithms in this study [48].

The foundation of the hierarchical analysis, the AHP involves the decomposition and synthesis of the relationships between the criteria until it reaches a prioritization of its indicators, approaching a better response of a single performance measurement [48].

The multi-criteria AHP method emerged in the late 1960s. It was developed by the mathematician Thomas L. Saaty when he was working for the US State Department's Arms Control and Disarmament Agency [49]. According to the authors, it was due to Saaty's observation about the difficulty of communication between members of the US government that the motivation to develop a decision-making support method arose.

The AHP method was developed to model unstructured problems in people's daily lives as they make decisions without necessarily having an exact notion of the parameters' importance [50].

It is currently applied to decision-making in several complex scenarios, where people work together to make decisions and in which human perceptions, judgments, and consequences have long-term repercussions [51]. This method is advantageous when the multi-criteria decision involves benefits, opportunities, costs, and risks [52].

The AHP method, in short, assumes that a set of criteria has been established and that it is trying to provide a normalized set of weights to be used when alternatives using criteria are being compared.

AHP involves three phases to solve the decision problem: decomposition, comparative judgments, and the synthesis of priorities.

First, we will address decomposition. The decomposition principle calls for the construction of a hierarchical network of criteria and alternatives to represent a decision problem, making it more easily analyzable and independently comparable, in which the global objective rests at the top, and the lowest levels represent the criteria and alternatives.

Second, there are the comparative judgments, in which, from the moment this logical hierarchy is built, decision-makers are asked to systematically evaluate the alternatives employing comparison, two by two, within each of the criteria, building a square comparison matrix. In this matrix, the decision-maker will represent, from a predefined scale of values that vary from 1 (indifference) to 9 (extreme preference) (Table 4) [52], his/her preference among the compared elements under the focus of an element of the immediately higher level. The comparison can use the factual data of the alternatives or human judgments as a form of underlying information [53].

**Table 4.** Comparative matrix A (assuming criterion 1 dominates criterion 2).

|  |  | Criterion 1 | Criterion 2 |
|---|---|---|---|
| **A =** | **Criterion 1** | 1 | Numerical Evaluation |
|  | **Criterion 2** | 1/numerical evaluation | 1 |

The main diagonal of the decision matrix is filled with a stipulated value to represent the non-dominance of one alternative over the other. If the $A_i$ element (from the row) is more critical than the $A_j$ element (from the column), any value from 1 to 9 must be entered (Table 5). Otherwise, if $A_i$ is less important than $A_j$, an inverse number to values 1 to 9 is inserted, i.e., 1/2, 1/3, and so on. In square matrices, we have $a_{ij}$, *para i = 1, 2, . . . , n e j = 1, 2, . . . , n*.

**Table 5.** SAATY's relative importance scale.

| Scale | Numerical Evaluation | Reciprocal |
|---|---|---|
| Extremely preferred | 9 | **1/9** |
| Between very strong and extremely | 8 | **1/8** |
| Very strongly preferred | 7 | **1/7** |
| Between strong and very strong | 6 | **1/6** |
| Strongly preferred | 5 | **1/5** |
| Between moderate and strong | 4 | **1/4** |
| Moderately preferred | 3 | **1/3** |
| Between equal and moderate | 2 | **1/2** |
| Equally preferred | 1 | 1 |

The total calculation of judgments for composing the comparison matrix for the pair is represented by n(n − 1)/2, corresponding to the number of judgments the decision-maker must make.

The collection of judgments is one of the fundamental steps for using the AHP. Simple and easy-to-understand mechanisms should be developed so that the evaluator (expert) can focus specifically on issuing judgments [54].

Priority synthesis is the last phase to solve a decision problem using AHP. Each alternative's composite weight must be calculated based on preferences derived from the comparison matrix in this stage. Saaty [48] explains that the determination of the priorities of the lowest factors concerning the objective is reduced to a sequence of comparisons by pairs between the levels, whether with feedback relationships or not. It is a rational way to deal with the judgments. Through these pairwise comparisons, the priorities calculated by the AHP capture subjective and objective measures and demonstrate the dominance intensity of one criterion over the other or one alternative over the other.

After assigning weights in the priority matrix, the relative weights must be calculated, such as the matrix normalization process, in which the priority vector is calculated [48]. Therefore, the result is the priority vector of the alternatives, that is, their order of importance. This vector will be multiplied by the criteria comparison matrix, generating an eigenvector. The eigenvector shows the dominance of each element concerning the others for a given criterion. An element not subject to a criterion receives the value zero in the eigenvector, not being computed in comparisons.

Next, it is necessary to identify the consistency of the matrices, exemplified by Hair et al. [42], with the reliability degree that a set of variables intends to measure. Saaty [48] claims that *A* is consistent if and only if $\lambda\ máx \geq n.$

$$\lambda max = \frac{1}{n} \sum_i \frac{A_{vi}}{Pi} \tag{9}$$

where $A_{vi}$ is eigenvector *i*, $P_i$ is priority vector *i*, and *n* is the number of alternatives.

To assess the closeness between $\lambda max$ and *n*, it is necessary to calculate the consistency ratio (*RC*), represented by the following formula:

$$RC = \frac{IC}{IR} \tag{10}$$

where **the** consistency index *(IC)* is represented by

$$IC = \frac{\lambda max - n}{n - 1} \tag{11}$$

According to [55–57], the AHP method can be summarized as the breakdown of a complex, unstructured situation into its parts. These parts, or variables, are arranged in hierarchical order; numerical values and subjective judgments are designated, denoting each variable's relative importance. Finally, judgments are synthesized to determine which variables have the highest priority and should be worked on to influence the situation's outcome.

Therefore, applying the AHP includes and measures all essential factors, qualitative and quantitatively measurable, tangible, or intangible, to approach a realistic model. However, bringing other approaches to the AHP methodology steps and their algebraic foundations is crucial.

In addition to presenting the classic AHP method, defined by Saaty in 1980 [50], an attempt was made to minimize the errors embedded in the classical method for obtaining the eigenvalues and the eigenvectors of the decision matrices. Hence, approximate algorithms, such as the normalized values method and the geometric mean method, were used, realizing that there are better decision-making results in complex scenarios. Therefore, the entire multi-criteria method discussed in this study was carried out using the geometric basis to minimize the errors generated by the classic AHP method and the mean normalization method for calculating the inconsistent matrix priority vector.

We will discuss two algorithms for obtaining the vector of priorities described in the literature for reciprocal and consistent matrices: approximate algorithms for calculating the eigenvector of consistent matrices, the average of normalized values, and the geometric average. In the following item, we address the AHP method with an average of normalized values as the first method of the approximate algorithms.

### 2.2.3. Analytic Hierarchy Process (AHP)—Average of Normalized Values Method

The average of the normalized values consists of the following steps:

(a) Normalization by the sum of each column's elements:

$$W_i(Mj) = \frac{a_{ij}}{\sum_{i=1}^{m} a_{ij}} j = 1, \ldots, n \tag{12}$$

(b)  The sum of elements of each normalized line, divided by order of the matrix:

$$W_i(M_i) = \sum_{j=1}^{m} W_i(Mj)/n \ \forall i = 1, \ldots, m \tag{13}$$

(c)  Calculation of the eigenvalue associated with the calculated vector in the previous item:

$$M \cdot W = \lambda max \cdot W \ \ \lambda max = \frac{1}{n} \sum_{i=1}^{n} \frac{[AW]_i}{wi} \tag{14}$$

The following item addresses the AHP method with the geometric mean:

2.2.4. Analytic Hierarchy Process (AHP)—Geometric Mean Method

The geometric mean method consists of the following steps based on a decision matrix:

(a)  The product of the elements of each row raised to the inverse of the order of the matrix:

$$Wj(M_i) = \sqrt[n]{\prod_{j=1}^{n} aij} i = 1, \ldots, n \tag{15}$$

(b)  Normalizing the obtained priority vector and calculating the eigenvalue associated with the calculated vector will produce an identical result to the λmax of the average normalized values method.

## 3. Results

This section presents the data and how the results were generated using hybrid algorithms. Built from machine learning algorithms (Random Forest, logistic regression, and Naïve Bayes) with a multi-criteria decision method (AHP) (resulting from the geometric mean), the hybrid algorithms identified the quality of mental health investigated during social distancing in the years 2020 and 2021 over the course of the COVID-19 pandemic. It also predicts the quality of mental health in pandemics that may occur in the future.

Tartuce 2006 points out that scientific methodology deals with method and science. Therefore, *method* (from the Greek *methods*, which means "mode of procedure") is the path toward a goal. On the other hand, *methodology* means the study of the method, a set of rules and procedures established to carry out scientific research derived from science, which contains a set of precise and methodically ordered knowledge concerning a specific domain of knowledge [58].

Therefore, scientific methodology is the systematic and logical study of the methods used in the sciences, their foundations, their validity, and their relationship with scientific theories. The scientific method generally comprises a set of initial data and a system of ordered operations suitable for formulating conclusions according to predetermined objectives [58].

Research is the main methodology action. It starts with a question, a doubt one wants to seek an answer. Therefore, research is meant to seek or find an answer to something. Human knowledge is characterized by the relationship established between the subject and the object, which is a relationship of appropriation. The complexity of the object to be known determines the scope of appropriation. Hence, the simple apprehension of everyday reality is widespread or empirical, while the in-depth and systematic study of reality is part of scientific knowledge [58,59].

By proposing hybrid algorithms built based on a multi-criteria decision method (AHP) (resulting from the geometric mean) with machine learning as a support for the analysis of the quality of mental health, the study fits as correlational and transversal with a quantitative approach. Being correlational, it aims to evaluate the relationship between two or more concepts, categories, or variables in a given context. The transversal factor of this study makes all the measurements in a single moment or during a short period, which is also useful in describing variables and their distribution patterns [60,61]. As for the means, the present research is a case study type, in which an individual case (phenomenon or

situation) is studied in depth to obtain an expanded understanding of other similar cases (phenomena or situations).

As for the procedures, the research is a survey that seeks information directly from an interest group about the desired data. It is a valuable procedure, especially in exploratory and descriptive research [62].

Survey research can obtain data or information about the characteristics or opinions of a specific group of people, indicated as representative of a target population, using a questionnaire as a research instrument [63].

Regarding the nature of the investigation, the present study can be classified as descriptive research. It aims to describe the characteristics of the study or establish relationships between variables through data collection using a questionnaire.

The proposed strategy used the multi-criteria method (AHP) based on the best result of the geometric mean. It can add to the criteria analysis concerning the diagnosis of mental health quality research during the years 2020 and 2021 in the period of social distancing due to the COVID-19 pandemic to generate initial weights that can be used in the training and testing of machine learning algorithms.

The principles of this research focus mainly on results from hybrid algorithms that unite the multi-criteria method and machine learning. This study uses supervised machine learning whose algorithms relate an output with an input based on labeled data. The labels are identified through classification, mapping input variables into different categories, such as the essential characteristics to be answered or qualified, in an evaluation that can assume predefined label sets. The classification also uses regression, predicting results in a continuous output [64].

A theoretical framework to analyze the consistency of the outputs within the framework is represented in Figure 1.

### 3.1. Method of Preparing the Database

It should be noted that on 17 March 2020, the Federal Institute of Piauí (IFPI), the object of this study, was marked with an unprecedented change since, given the confirmation of the global COVID-19 pandemic, through Regulatory Ordinance nº 853 [65], it suspended its in-person classroom activities. Therefore, this research includes students and employees of the Federal Institute of Education, Science, and Technology of Piauí (IFPI) who experienced social distancing during the COVID-19 pandemic in 2020 and 2021.

The database was extracted from a cross-sectional and quantitative study of an exploratory nature carried out with individuals aged 13 to 65 years old in the years 2020 and 2021. This study results from research that is part of the REASSESSMENT REPORT ON THE IMPACTS OF COVID-19 PANDEMIC ON IFPI STUDENTS AND EMPLOYEES, which sought to investigate the impacts of social distancing on the mental health of students and employees of the Federal Institute of Piauí (IFPI). It is a multi-curriculum and multi-campus institution that has, in all units, a total of 2230 employees and 23,396 students among those regularly enrolled and those who are still enrolling, in addition to 560 outsourced employees.

The study was conducted in the state of Piauí, located in the northeast of Brazil (See Figure 2). The research cities went beyond the state capital, Teresina; some other urban cities, such as Parnaíba, Floriano, Piripiri, Picos, Pedro II, Corrente, São Raimundo Nonato, Oeiras, Campo Maior, Uruçuí, Angical, Cocal, Valença, and São João do Piauí, are all far from the capital and located throughout Piauí in the north, south, and southeast of the state.

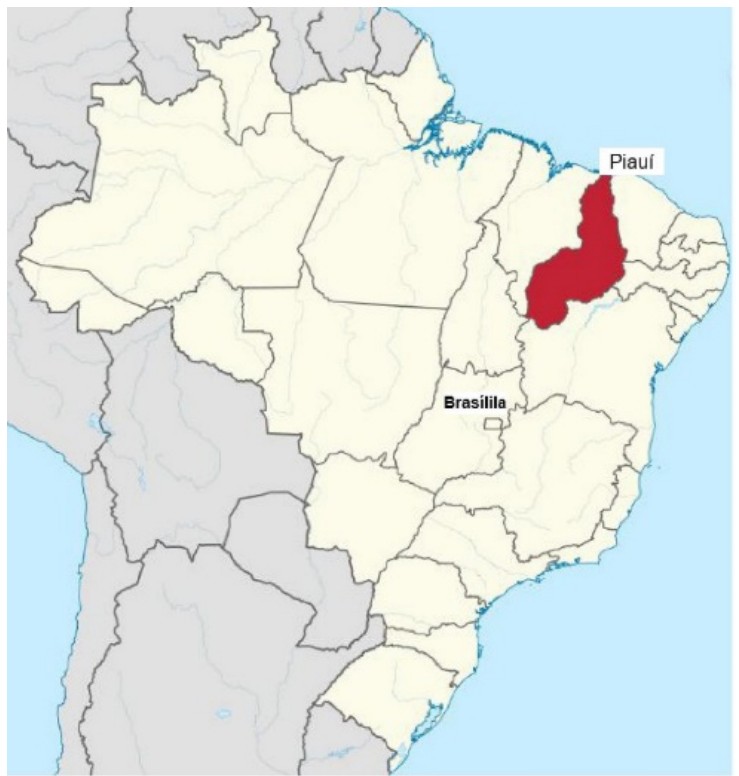

**Figure 2.** A map of Brazil highlighting the State of Piaui and Brasília.

What was the number of COVID-19 infections and deaths in Brazil and the study area during the study period? Is it possible to show this information graphically (Shown in Figures 3 and 4).

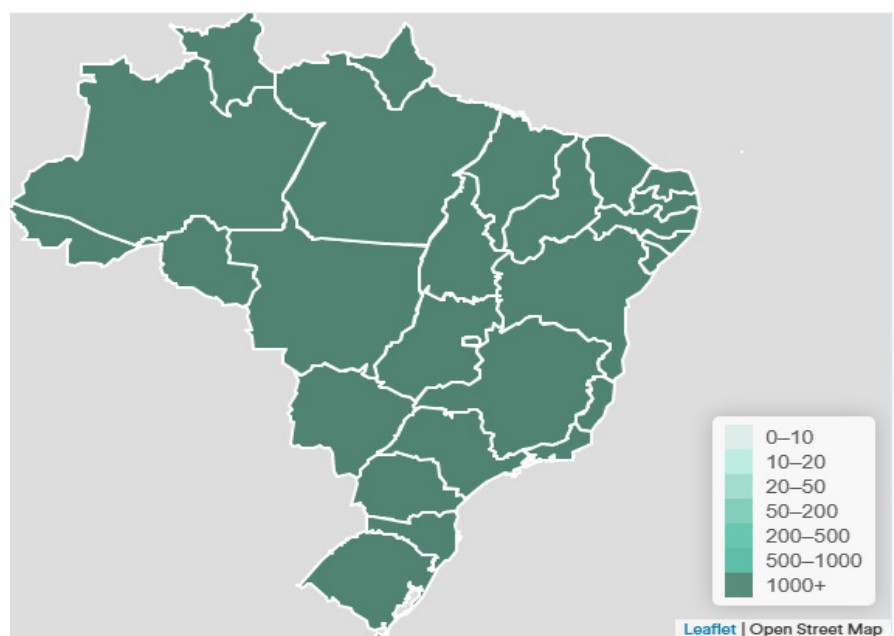

**Figure 3.** COVID-19 incidence coefficient by the state of notification, 2020.

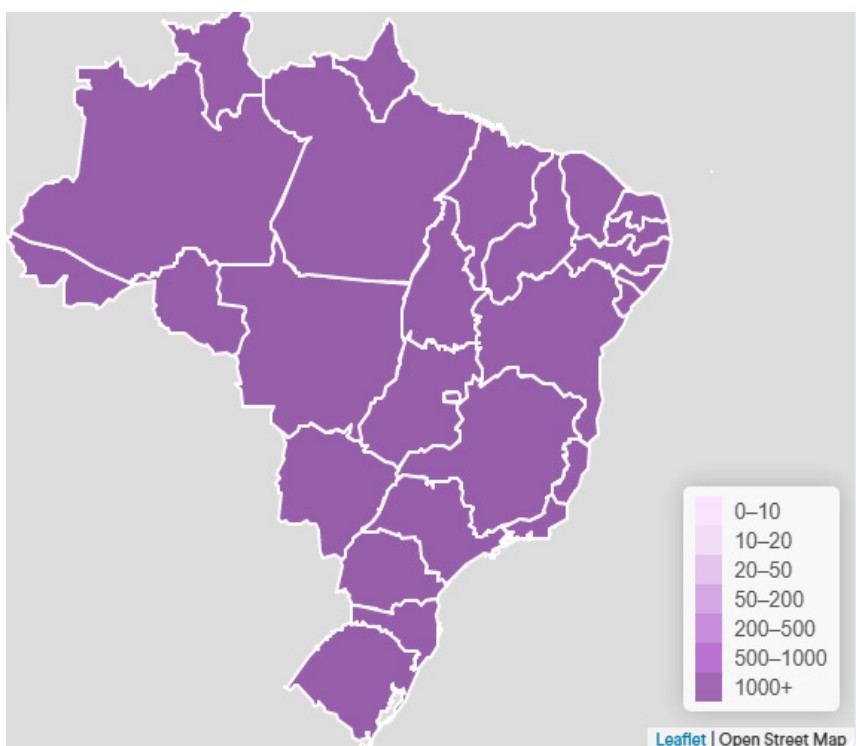

**Figure 4.** Deaths from COVID-19 by the state of notification, 2020.

The subjects were asked through their institutional email to participate in the study and answer an online questionnaire prepared in Google Forms through the electronic address https://forms.gle/tJHJMNRyxzn5cmMR8 (accessed on 31 March 2020). The first data collection was carried out in June 2020, and the first Google Forms questionnaire could be answered until 12 April 2020, with the exception that the answers would not be identified and the respondent's email would not be collected. Psychologists from the Federal Institute of Piauí coordinated the research. For the IFPI campus, 3255 questionnaires were answered online.

In March and April 2021, due to the extension of the COVID-19 pandemic and the maintenance of social distancing, a new survey was also carried out through an online questionnaire prepared in Google Forms, again with IFPI students and employees following the same criteria as the first data collection, with a total of 2168 respondents.

As a form of evaluation in 2020 and to reassess the aspects related to the mental health of the IFPI school community, in 2021, the research aimed to assess the mental health vulnerability points to which the IFPI students and staff were exposed during social distancing during the COVID-19 pandemic. The online questionnaires, prepared on Google Forms, applied in 2020 and 2021, addressed questions about emotions, feelings, and behaviors experienced due to social distancing during the COVID-19 pandemic in 2020 and its extension in 2021. They also addressed the possible changes that could have occurred in various aspects, such as mood, self-confidence, interest in life, eating and sleeping habits, family life, and interest in working/studying.

The data treatment was carried out through content analysis. The results show that, for researchers, the leading causes of confirmation bias are related to the desire to succeed and the issue of some topics being more controversial.

The extension of the pandemic period and the uncertainties regarding the return of classroom activities impacted the school community's physical and emotional health. They put individuals at risk of developing psychopathological manifestations. In addition to the psychological implications directly related to the disease, measures to contain the pandemic may have jeopardized mental health. Some research has already reported the harmful

effects of prolonged social isolation, such as symptoms of post-traumatic stress, confusion, and anger [66] and concerns about shortages of supplies and financial losses [67].

Next, data from the study profile in the survey carried out at IFPI will be presented.

First, we have graphs on gender in the surveys of the years 2020 and 2021, presented in Figure 5.

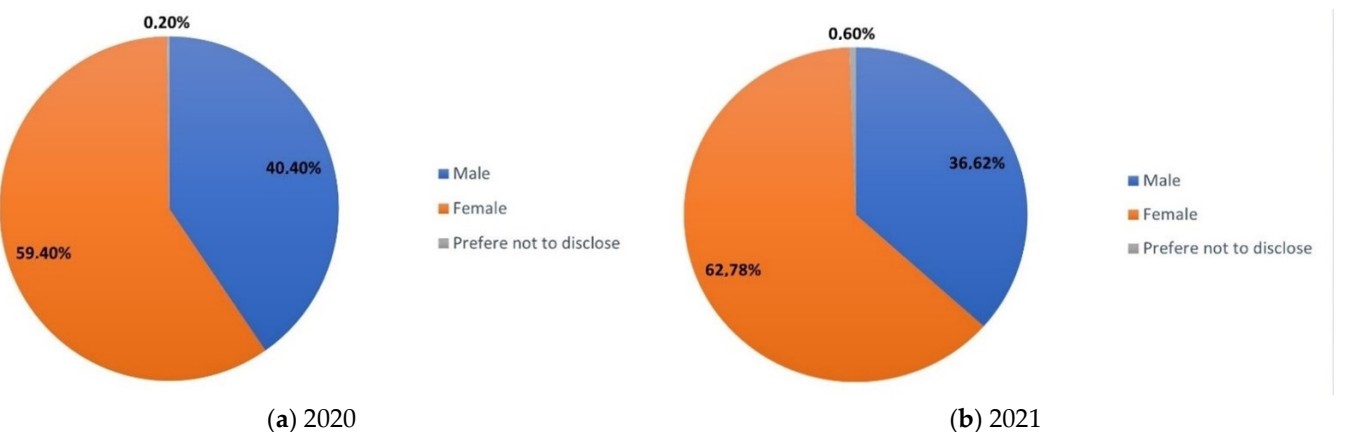

(**a**) 2020                                                                                    (**b**) 2021

**Figure 5.** Graphs on gender: (**a**) 2020; (**b**) 2021.

In the 2020 and 2021 surveys, respondents verified possible changes in mental health (*for the worse*, *for the better*, or *without change*) that could have occurred for each of the four study criteria. The objective was to identify mental health quality during social distancing during the COVID-19 pandemic, as seen in Table 6. Below, there is a description of the criteria.

**Table 6.** Criteria—2020 and 2021.

| Critérios | Definições dos Critérios |
| --- | --- |
| Cr1: Emotions and feelings | Related aspects to possible changes in emotions and feelings |
| Cr2: Physical health | Related aspects to possible changes in physical health |
| Cr3: Interpersonal relationships | Related aspects to possible changes in interpersonal relationships |
| Cr4: Routine | Related aspects to possible changes in daily behavioral routine |

Table 7 contains the alternatives of the possible questionnaire answers for the years 2020 and 2021.

**Table 7.** Alternatives—2020 and 2021.

| Alternatives |
| --- |
| a1: Worse than before |
| a2: No change |
| a3: Better than before |

First, the following questions were asked in the online questionnaire on the emotions and feelings criterion during social distancing due to the COVID-19 pandemic in the years 2020 and 2021:

- Your mood;
- Your self-confidence;
- Your interest in life;
- Your ability to endure difficult situations.

Afterward, subsequent questions on the physical health criterion during social distancing due to the COVID-19 pandemic in the years 2020 and 2021 were asked in the online questionnaire:

- Your eating habits;
- Your energy (willingness to do things);
- Your sleep;
- Your physical health (pain, tremors, malaise);
- Your sexuality (sexual satisfaction).

Then, the following questions on the interpersonal relations criterion during social distancing due to the COVID-19 pandemic in the years 2020 and 2021 were asked in the online questionnaire:

- Your coexistence with your family (the one you live with);
- Your coexistence with friends;
- Your coexistence with other people;
- Your financial conditions for family support.

Finally, the questions below were asked in the online questionnaire on the routine criterion during social distancing caused by the COVID-19 pandemic in the years 2020 and 2021:

- Your interest in working/studying;
- Your leisure activities (the things you like to do);
- Your ability to fulfill obligations;
- Your household tasks (cooking, cleaning the house, shopping, fixing things);
- Your interest in engaging in other activities.

### 3.2. Description of the Proposed Hybrid Model Development

First, in the informative phase, a survey was carried out to learn which studies were conducted during social distancing caused by the COVID-19 pandemic in 2020 and 2021 in the mental health quality field to know whether computational means were used in these studies to quantify the current and future situation.

Thus, a literature review was carried out where articles published in journals, books, theses, dissertations, and conference proceedings were observed through a search on the Web of Science (WOS) indexing database. The same query was carried out on the indexing database SCOPUS. To be included in the review, the title or keywords of an article needed to contain the sequence ("mental health") AND ("COVID-19") AND ("social distancing") AND ("AHP") AND ("MACHINE LEARNING"). The terms were connected through Boolean logic, a method by which the database can search for specific combinations of keywords. As an additional inclusion criterion, the publication type was defined as a journal article in English and published between 2020 and 2023.

A search was carried out on the Web of Science (WoS) indexing database, which resulted in 1582 published articles after applying only the keywords ("mental health") AND ("COVID-19") AND ("social distancing"). The same query was performed on the SCOPUS indexing database, returning 1014 published articles. When refining the search, the terms AHP and MACHINE LEARNING were included in the bibliographic review. The result was reduced to 378 articles in the Web of Science (WoS) and 109 articles in SCOPUS in Portuguese and English. The results of this scientific search gave direction to this study, as it is an unprecedented subject in renowned scientific databases, highlighting the importance of continuing the work [68–70].

With the information mentioned in item 3.1, a fundamentally important database for training and testing the proposed hybrid algorithms was generated in a second step. The database refers to 2020 and 2021, corresponding to 70% for training and 30% for testing.

Subsequently, it was applied in the training and testing stages of the machine learning algorithms (Random Forest, logistic regression, and Naïve Bayes) and, later, the multi-criteria decision-making method AHP resulting from the geometric mean. These algorithms

have been successfully implemented in several fields, mainly in health, for classification and regression purposes.

The computational tool Orange Canvas Framework (version 3.32.0) was used to implement the machine learning algorithms. The Orange Canvas is an open-source tool that uses a library in the Python scripting language and visual programming. This library was designed to simplify data analysis workflows, and design data mining approaches from a combination of existing components (widgets).

Next, hybrid algorithms and their results will be addressed.

Hybrid Algorithms

The first stage of problem modeling and, consequently, the hybrid algorithms was the investigative phase, in which an investigation was conducted on the research that was to be carried out. In the second stage, a search was performed on the multi-criteria methods and machine learning algorithms. In the third stage, a literature review was carried out to identify the main themes of the research. In the fourth phase, online questionnaires were conducted in 2020 and 2021, which sought to identify the mental health quality of Federal Institute of Piauí (IFPI) students and employees during the social distancing caused by the COVID-19 pandemic, as described in item 3.1 of this study.

In the fifth modeling phase, also called the structuring stage, the data were preprocessed to identify the classifications of the characteristics, extract knowledge, and evaluate them through a scientific investigation; their quality can compromise the interpretation of the satisfactory result of the proposed models. Subsequently, the hybrid machine learning algorithms (Random Forest, logistic regression, and Naïve Bayes) with a multi-criteria decision-making method, AHP, resulting from the geometric mean, were elaborated in the same phase. Finally, the computational tool Orange Canvas was used for data processing and visualization of the performance of learning algorithms.

Figure 6 shows the modeling problem addressed in this study using the Orange Canvas software.

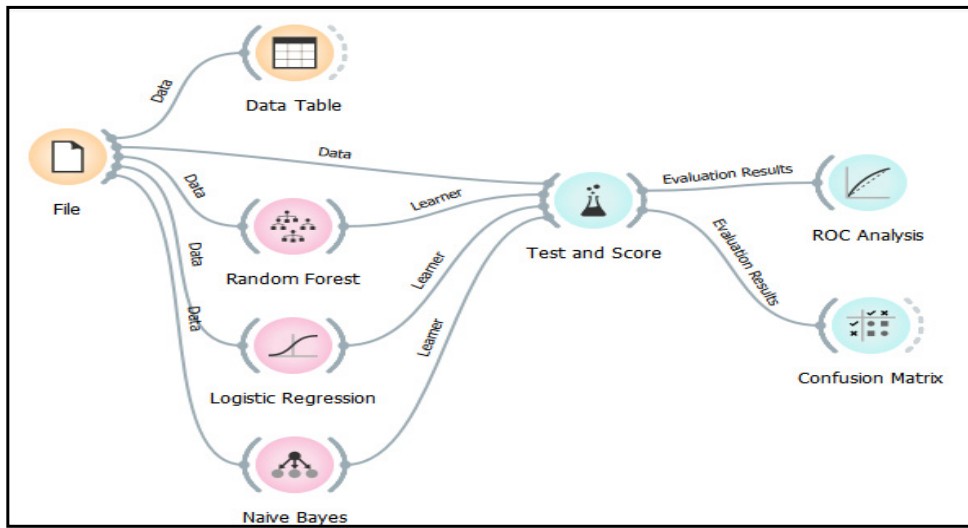

**Figure 6.** Modeling problem using machine learning—Orange Canvas.

Ultimately, the sixth and last phase consisted of the proposed data analysis; the results will be discussed later. Below, the author prepared Figure 7 of the proposed research methodology according to the steps described.

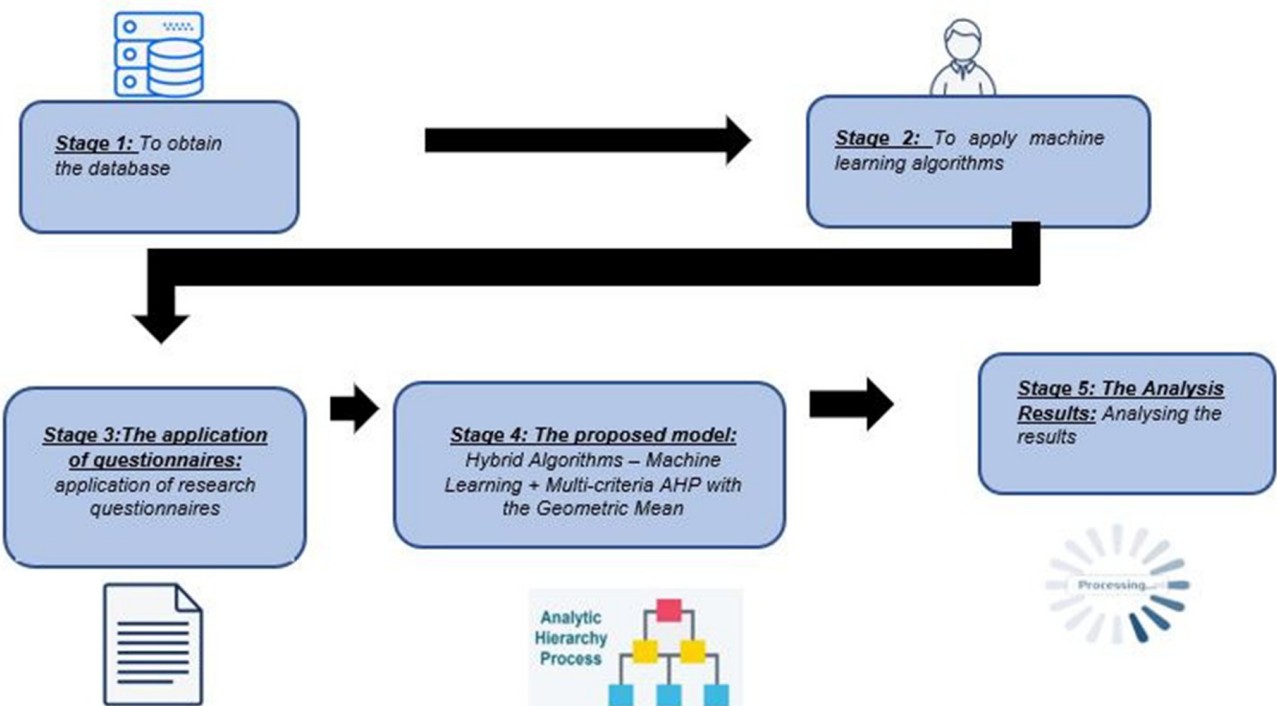

**Figure 7.** Formatted by the author.

The origin of the test base for each cycle involved exploring a large and varied amount of data structured in rows and columns. The learning cycle had to be repeated until the results were satisfactory, making the algorithm more accurate.

In the 2020 database, there are 3255 samples, of which 70% were used for training and 30% for testing, and in the 2021 database, there are 2168 samples, of which 70% were used for training and 30% for testing. When using the Orange Canvas software, the NO CHANGE alternative was selected as a target in the FILE widget. It was more relevant after creating the initial weights with the AHP multi-criteria method with the geometric mean.

The initial preprocessing flow in the Orange Canvas software had the following input data concerning the year 2020: 980 instances, 120 variables, and three features; in the year 2021, there were 650 instances, 120 variables, and three features.

The starting point of the methodology presented in this study was the elaboration phase of the hybrid machine learning algorithms (Random Forest [71], logistic regression [72], and Naïve Bayes [73]) with a multi-criteria decision method [74], AHP, resulting from the geometric mean [75].

In the 2020 and 2021 questionnaires, the survey respondents were asked to verify possible changes concerning alternatives (worse than before, no change, or better than before) that could have occurred in each of the items listed in the collected answers.

Figure 8 represents the workflow in Orange Canvas, initially holding data from 2020. The same was carried out for the year 2021.

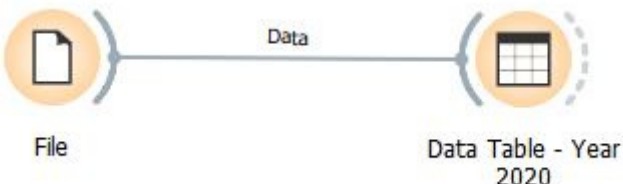

**Figure 8.** Data initial preprocessing flow.

For the training and test base, k-fold cross-validation (with k = 10) was applied. We performed the adaptation procedure ten times, and each adaptation was performed in a training session.

The machine learning algorithm (Random Forest, logistic regression, and Naïve Bayes) was processed for predictive characteristics of the situation's high risk in the face of the emotions and feelings criterion, which performed better with the NO CHANGE alternative in the year 2020.

The importance judgments between pairs of criteria (paired judgments) were used concerning the main focus in applying the AHP multi-criteria method to improve the results of the machine learning algorithms. Therefore, the value scale established by Saaty [45] was considered.

The results of these judgments are presented in Tables 8–11, referring to 2020. Continuing the application of the AHP method in aiding the proposed decision, in the elaboration of the hybrid machine learning algorithms for the year 2020, the tables show the analysis of each of the criteria, which were compared pair by pair (parity judgments). After this evaluation, it was possible to estimate from the geometric mean of the criteria comparison matrix lines, resulting in the EIGENVECTOR column (weighting eigenvector) in the classic multi-criteria AHP through the average of normalized values and the geometric average.

**Table 8.** AutoVetor—2020.

| | AUTOVETOR 2020 | | | | | | | | | | | |
| | AHP Classic | | | | AHP Normalized Values | | | | AHP with Geometric Mean | | | |
| | CR1 | CR2 | CR3 | CR4 | CR1 | CR2 | CR3 | CR4 | CR1 | CR2 | CR3 | CR4 |
|---|---|---|---|---|---|---|---|---|---|---|---|---|
| WORSE THAN BEFORE (%) | 0.30 | 0.28 | 0.32 | 0.34 | 0.32 | 0.28 | 0.32 | 0.34 | 0.28 | 0.32 | 0.34 | 0.34 |
| Same as BEFORE (%) | 0.48 | 0.42 | 0.44 | 0.46 | 0.46 | 0.42 | 0.44 | 0.46 | 0.42 | 0.44 | 0.46 | 0.43 |
| Better THAN BEFORE | 0.21 | 0.20 | 0.14 | 0.12 | 0.27 | 0.20 | 0.14 | 0.12 | 0.20 | 0.14 | 0.12 | 0.13 |

**Table 9.** Table of indexes and consistency ratio—2020.

| | Consistency Index Results—The Year 2020 | | | |
| Index/Alternatives | Criterion: CR1: Emotions and Feelings | Criterion: CR2: Physical Health | Criterion: CR3: Interpersonal Relationships | Criterion: CR4: Routine |
|---|---|---|---|---|
| IC | 0.18 | 0.22 | 0.29 | 0.33 |

**Table 10.** Eigenvector in 2021.

| | Eigenvector 2021 | | | | | | | | | | | |
| | AHP Classic | | | | AHP Normalized Values | | | | AHP with Geometric Mean | | | |
| | CR1 | CR2 | CR3 | CR4 | CR1 | CR2 | CR3 | CR4 | CR1 | CR2 | CR3 | CR4 |
|---|---|---|---|---|---|---|---|---|---|---|---|---|
| WORSE THAN before (%) | 0.34 | 0.56 | 0.03 | 0.25 | 0.56 | 0.56 | 0.03 | 0.25 | 0.56 | 0.03 | 0.25 | 0.37 |
| SAME AS BEFORE (%) | 0.43 | 0.35 | 0.46 | 0.53 | 0.35 | 0.35 | 0.46 | 0.53 | 0.35 | 0.46 | 0.53 | 0.40 |
| Better THAN BEFORE | 0.13 | 0.06 | 0.09 | 0.10 | 0.06 | 0.06 | 0.09 | 0.10 | 0.06 | 0.09 | 0.10 | 0.14 |

**Table 11.** Table of indexes and consistency ratio—2021.

| | Consistency Index Results—The Year 2021 | | | |
|---|---|---|---|---|
| **Index/Alternatives** | **Criterion: CR1: Emotions and Feelings** | **Criterion: CR2: Physical Health** | **Criterion: CR3: Interpersonal Relationships** | **Criterion: CR4: Routine** |
| *IC* | 0.18 | 0.23 | 0.27 | 0.32 |

Table 8 deals with the parity judgments by the emotions and feelings criterion concerning the alternatives. To include the EIGENVECTOR column (weighting eigenvector), we used the classic AHP method, the average of normalized values, and the geometric mean.

CR1 corresponds to the emotions and feelings criterion; CR2 is the physical health criterion; CR3 is the interpersonal relationships criterion; and CR4 is the routine criterion.

In Table 9, the consistency index (CI) calculation results are performed, resulting in the values presented.

In Table 10, CR1 corresponds to the emotions and feelings criterion; CR2 is the physical health criterion; CR3 is the interpersonal relationships criterion; and CR4 is the routine criterion. To include the eigenvector column (weighting eigenvector), we used the classic AHP method, the average of normalized values, and the geometric mean.

The confusion matrix presents options 1.0 (BETTER THAN BEFORE), 2.0 (SAME AS BEFORE), and 3.0 (WORSE THAN BEFORE) regarding the emotions and feelings characteristics of Figures 5 and 6. Regarding the trained and tested algorithms, it obtained the total instances of 2.0 (SAME AS BEFORE) using the Random Forest algorithm in 2020 and 2021.

The results of the machine learning algorithms' performance presented in Table 12 were able to identify which emotions and feelings stood out in all situations brought to the research through the AHP method with the geometric mean. The result remains the SAME AS BEFORE in 2020.

**Table 12.** Learning algorithms performance table—2020.

| Model | Train Time (s) | Test Time (s) | AUC | CA | F1 | Precision | Recall | Log Loss | Specificity |
|---|---|---|---|---|---|---|---|---|---|
| Random Forest | 0.034 | 0.004 | 0.999 | 0.980 | 0.970 | 0.980 | 0.960 | 0.044 | 0.990 |
| Naïve Bayes | 0.007 | 0.008 | 0.975 | 0.887 | 0.828 | 0.837 | 0.820 | 0.183 | 0.920 |
| Logistic Regression | 0.042 | 0.001 | 0.997 | 0.973 | 0.959 | 0.940 | 0.940 | 0.120 | 0.990 |

The hybrid model using machine learning with the Random Forest algorithm had the best performance due to obtaining an accuracy (column AC) of 98.0%, compared with a logistic regression accuracy of 97.3% and a Naïve Bayes accuracy of 88.7%. Precision indicates the number of correct choices across all class classifications for the high risk that mental health follows the SAME AS BEFORE alternative. Recall indicates one of the high-risk situations, and F1 reveals the harmonic mean between precision and recall, as seen in Table 12, in addition to informing the training and testing time used by Orange Canvas.

Another relevant metric to this research is the confusion matrix, which can reveal the number of correct and incorrect instances classified by the algorithm referring to the year 2021 in a total of 980 instances, where 1.0 represents WORSE THAN BEFORE, 2.0 represents SAME AS BEFORE, and 3.0 represents BETTER THAN BEFORE, symbolized in Figure 9.

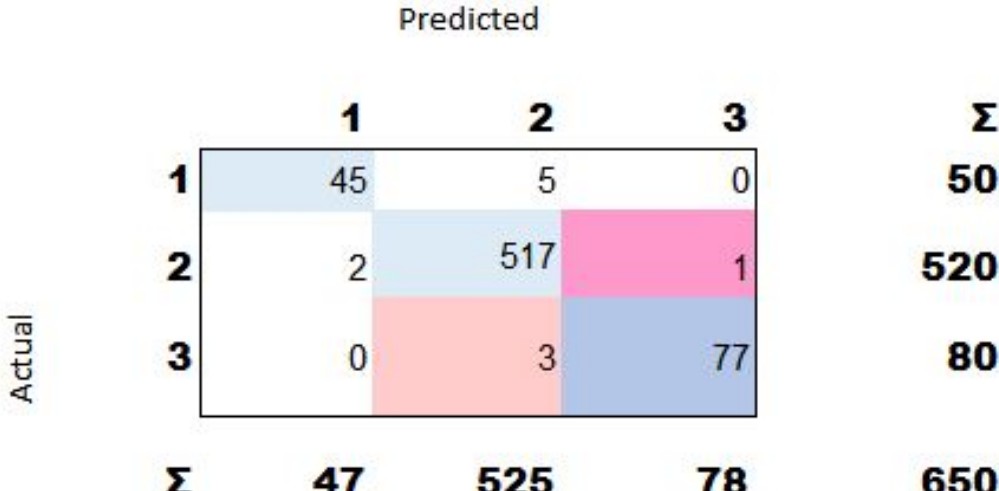

**Figure 9.** Classification accuracy number of Random Forest instances—2021.

For the results found in 2021, the problem modeling was used by from the Orange Canvas software (1.0). Similar to 2020, the construction of hybrid algorithms refers to processing 70% of the data for training and 30% for testing, using the initial weights generated through the AHP multi-criteria decision method.

In Table 13, showing the performance of the machine learning algorithms, the results identified which emotions and feelings stand out in all situations brought to the research through the AHP method with their initial weights, and the development WORSE THAN BEFORE was obtained in the year 2021. The result was analyzed using the ACURA value (AUC column).

**Table 13.** Learning algorithms performance—2021.

| Model | Train Time (s) | Test Time (s) | AUC | CA | F1 | Precision | Recall | Log Loss | Specificity |
|---|---|---|---|---|---|---|---|---|---|
| Random Forest | 0.028 | 0.004 | 1.000 | 0.995 | 0.995 | 0.995 | 0.995 | 0.044 | 0.990 |
| Naïve Bayes | 0.009 | 0.001 | 0.992 | 0.911 | 0.915 | 0.931 | 0.911 | 0.339 | 0.972 |
| Logistic Regression | 0.046 | 0.001 | 0.971 | 0.895 | 0.883 | 0.889 | 0.895 | 0.284 | 0.841 |

The confusion matrix reveals the number of correct and incorrect instances classified by the algorithm referring to the year 2021 in a total of 980 instances, where 1.0 represents WORSE THAN BEFORE, 2.0 represents SAME AS BEFORE, and 3.0 represents BETTER THAN BEFORE, as depicted in Figure 10.

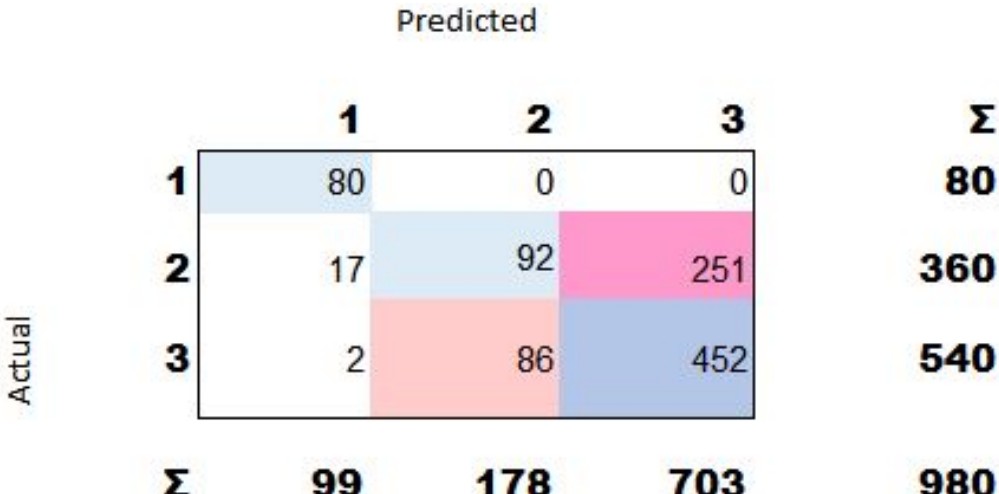

**Figure 10.** Classification accuracy number of Random Forest instances—2021.

## 4. Discussion

To identify mental health quality during the social distancing caused by the COVID-19 pandemic in the years 2020 and 2021, the following discussions of the obtained results in the previous section are presented.

The study presents some limitations regarding the scope and type of sample, but it is possible to generalize the results found here beyond the participants of this study. Generalizations are possible for the rest of the state, country, and world, as well as for social groups other than those presented here.

It should be noted that the State of Piauí, as well as other regions of Brazil and other countries, suffered economic, financial, and mainly psychological damage to its population. Considering that the physical and mental health of a population (be it in Piauí (Brazil) or any other country) is influenced by the socioeconomic context, the labor market, social assistance, and public policies, it is possible to consider that changes in these dimensions reflect the indicators of well-being, especially in the event of mental disorders.

The machine learning algorithms' performance identifies the result SAME AS BEFORE in 2020 as the best result for the semotions and feelings characteristics. The hybrid model using machine learning with the Random Forest algorithm had the best performance of accuracy (AUC) of 98.0%, compared with a logistic regression of 99.7% and a Naïve Bayes of 97.50%, in the year 2020.

On the other hand, the machine learning algorithms' performance identifies the result WORSE THAN BEFORE in 2021 as the best result for the emotions and feelings characteristics. The hybrid model using machine learning with the Random Forest algorithm had the best performance of accuracy (AUC) of 100%, compared with a Naïve Bayes of 91.1% and a logistic regression of 89.50%, in 2021.

This demonstrates the best hybrid algorithm corresponding to Random Forest with AHP with a geometric average. The emotions and feelings criterion in the year 2021 was found during the training samples and tests. This presented the most significant number of samples in terms of result 3.0 (WORSE THAN BEFORE) in 2021 and result 2.0 (SAME AS BEFORE) in 2020.

The confusion matrix shows 1.0 (BETTER THAN BEFORE), 2.0 (SAME THAN BEFORE), and 3.0 (WORSE THAN BEFORE) regarding the emotions and feelings characteristics of Figures 5 and 6. Regarding the trained and tested algorithms, it obtained the total instances of 2.0 (SAME AS BEFORE) with the Random Forest algorithm in 2020 and 2021.

Figure 5 demonstrates the classification accuracy of the hybrid algorithms with Random Forest in 2020 by using the confusion matrix, which revealed the number of instances correctly classified by the machine learning algorithm. Regarding the trained and tested

algorithms, it obtained a precision of 517 samples for 2.0 (SAME AS BEFORE) using the Random Forest algorithm.

Figure 6 demonstrates the classification accuracy of the hybrid algorithms with Random Forest in the year 2021 using the Confusion Matrix, which revealed the number of instances correctly classified by the machine learning algorithm. Regarding the trained and tested algorithms, it obtained a precision of 452 samples for 3.0 (WORSE THAN BEFORE) using the Random Forest algorithm.

Concerning the methodology would be, in the future, researchers should consider carrying out tests with other machine learning algorithms or other alternatives, such as an artificial neural network, which could predict criteria using different databases related to other aspects of the diagnosis of mental health.

The use of specific controls to validate the insertion of a new methodology should also be considered. In addition, the correlation between previous research and new research is useful in knowing the real contribution to mental health diagnosis.

In addition, Figure 8 shows the classification accuracy of the hybrid algorithms with logistic regression, Naïve Bayes, and Random Forest in the year 2020 using the confusion matrix, which is capable of revealing the number of instances correctly classified by the machine learning algorithm. The results of 1.0 (WORSE THAN BEFORE), 2.0 (SAME AS BEFORE), and 3.0 (BETTER THAN BEFORE) represent the emotions and feelings characteristics. Regarding the trained and tested algorithms, we obtained the total instances of 2.0 (SAME AS BEFORE) with the Random Forest algorithm using 452 samples.

Furthermore, Figure 10 demonstrates the classification accuracy of the hybrid algorithms with logistic regression, Naïve Bayes, and Random Forest in 2021 using the confusion matrix, which revealed the number of correct instances classified by the machine learning algorithm. Regarding the trained and tested algorithms, it obtained a precision of 517 samples of 2.0 (WORSE THAN BEFORE) using the Random Forest algorithm.

The study used hybrid algorithms with a combination of machine learning algorithms and the AHP multicriteria method to diagnose mental health using a database applied to an educational institution that saw its educational activities go from the in-person teaching model to remote teaching.

This research was carried out to evaluate the points of vulnerability to which students and employees were exposed and to collect data that could support the planning of mental health actions in an educational institution. This survey made it possible to compare the institution's situation at two significant moments, the initial period of the COVID-19 pandemic and one year into the pandemic and point out the different perceptions experienced by students and employees about their mental health.

In a more generalized analysis, it was noticed that there was an increase in the indices that demarcate a worsening of symptomatic conditions concerning mental health, the focus of this survey. In addition, the study can serve as an instrument for effective and efficient actions in meeting the demands perceived throughout the study and to understand how the post-pandemic is a time to incorporate the knowledge acquired during this study, using the information for public policies in the educational field related to mental health. It will be essential to address the emotional needs of the actors in this process of gradually leaving the pandemic.

## 5. Conclusions and Future Studies

The theme of this study is original because it deals with hybrid algorithms involving machine learning technology combined with the AHP multicriteria method. It has relevance in the application area because it deals with a technique never before applied during a pandemic and is not used for mental health diagnoses during social distancing.

The study manages to fill a gap in hybrid algorithms because it combines machine learning and AHP to deal with an unprecedented event, a study during a pandemic.

As for the main contributions of the results obtained in this study, it is possible to perceive that a hybrid algorithm (machine learning with the AHP multicriteria method)

can be used as an alternative for the identification of criteria that are more likely to be related to the diagnosis of mental health during the COVID-19 pandemic due to social distancing, which can help in the decision-making process regarding public health planning and intervention in this area [76–78].

Identifying the quality of mental health during the social distancing caused by the COVID-19 pandemic in the years 2020 and 2021 was a complex process, given the questions experienced by each respondent in the survey's online questionnaires.

The hybrid approach of the machine learning algorithm with the AHP multi-criteria decision method with the geometric mean accurately obtained a classification that stood out the most among the characteristics concerning emotions and feelings. In 2020, the situation was reported as the SAME AS BEFORE, in which the hybrid AHP with the geographical average with machine learning Random Forest Algorithm stands out, highlighting the atypical situation in the interviewees' quality of life and the amount of time it takes to realize that mental health remains unchanged.

In 2021, the situation was reported as WORSE THAN BEFORE, in which the hybrid AHP with the geometric mean with machine learning Random Forest Algorithm provided an absolute result. After more than 365 days of the COVID-19 pandemic and maintaining social distancing as the primary strategy in dealing with the pandemic, the worsening situation in the quality of mental health is highlighted.

New predictive models of machine learning are expected to be developed using multi-criteria AHP with processing by obtaining the eigenvalues and eigenvectors of the decision matrices and using the direct eigenvector method as a way of comparing the results of the algebraic and numerical methods. In that case, it will be possible to compare the results of the normalization methods, mainly with the geometric average used in the current study.

As for the social contribution, this study can potentially help public policies in educational institutions, such as the Federal Institute of Piauí. It also assists with society's mental health, identifying its quality during social distancing in pandemics that may occur in the future that require social distancing.

The results are encouraging for future studies of criteria analysis that show the quality of mental health over the course of social distancing experienced during the COVID-19 pandemic. It is expected that we can predict situations regarding mental health with other machine learning algorithms, such as artificial neural networks.

On that note, there are indications of new experiments using other databases, applying the same methodology used in this work, and developed for the comparative analysis of results. This presents the possibility for future studies.

Therefore, since new outbreaks of viruses may occur worldwide, and viral epidemics can have a psychosocial impact, services in mental welfare, development, support, treatment, and mental health assessment are crucial. These are urgent goals in implementing collective health in 2023 and other future pandemics through action planning and public policies.

Concerning the methodology, in the future, we should carry out tests with other machine learning algorithms or other alternatives, such as an artificial neural network, which could work with the prediction of criteria using different databases related to other aspects of mental health diagnosis. The use of specific controls to validate the insertion of a new methodology and the correlation between previous research and further research could be considered to know the real contribution to mental health diagnoses [79].

**Author Contributions:** All authors contributed equally to the preparation of this article. W.S.C. and N.M.d.S. were responsible for the research, collecting data, and applying computational techniques, and relied on the help of P.R.P. and L.d.A.F.C. for analysis and the conclusions of the results. All authors have read and approved the final version of this article. All authors have read and agreed to the published version of the manuscript.

**Funding:** This research was funded by the National Council for Scientific and Technological Development (CNPq), grant number 304272/2020-5.

**Institutional Review Board Statement:** Not applicable.

**Informed Consent Statement:** Not applicable.

**Data Availability Statement:** No new data were created.

**Acknowledgments:** The first author is a federal public servant at the Federal Institute of the State of Piauí, Brazil, and a doctorate student in the research field of applied informatics at the University of Fortaleza (UNIFOR). The second author would like to thank Fundação Edson Queiroz/Universidade de Fortaleza and the National Council for Scientific and Technological Development (CNPq).

**Conflicts of Interest:** The authors declare no conflict of interest.

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
