# Peer review of "Aligning the Goals Hybrid Model for the Diagnosis of Mental Health Quality"

_sustainability, doi:10.3390/su15075938_

Round 1

Reviewer 1 Report

Please improve the structure of the paper. A clear method chapter is needed. How the respondents have been selected. Any bias ? 

A clear description of the role of machine learning associated with the main focus of the article is needed. Please develop a theoretical framework to be able to analyze the consistency of the outputs with the framework. 

A state of the art of the literature  is needed. 

The paper needs to have clear blocks of knowledge ,introduction, theoretical framework, method, results and discussion.  Please indicate how the outcomes of the study has been validated by whom with whom etc 

Author Response

Dear Reviewer,

With this letter of reply, we thank you for the comments and suggestions that helped revise the work, which certainly qualifies and improves our text. We inform you that the changes suggested by the reviewer were fulfilled, answered, and changed in the article. Also, the changes in the text of the article are in the font in blue so that it is better to visualize. The following are the responses to the reviewer's comments and suggestions.

Sincerely,

The authors

Brazil

Reviewer 2 Report

Images must be revised. Their quality is no good

Author Response

(The authors gave the same response as above.)

Reviewer 3 Report

This manuscript addresses social distancing in the COVID-19 pandemic. The main findings of this paper are important in fully understanding the social changes brought about by the COVID-19 pandemic. However, the authors need to state whether the model can be adapted to all countries, not just Brazil, which is the subject of this study. Overall, I would suggest extensive revision in combination with re-review for this manuscript.

[2. Problem Definition and Optimization Model] Is the area where this study was conducted far from the Brazilian capital? Or are they located in urban areas? If possible, I would like you to indicate this on a map. Also, what was the number of COVID-19 infections and deaths in Brazil and the study area during the study period? Is it possible to show this information graphically?

[4. Discussion] Were there any limitations in this study? What are some of the problems that could be solved in order to apply this research to the real world?

Author Response

(The authors gave the same response as above.)

Reviewer 4 Report

1. What is the main question addressed by the research?

First, authors should describe contribution and significance of this study. Second, authors should improve the structure and content of this study. Third, the method/research design should be detailed described, such as respondents select strategy. Finally, discussion should be detailed described.

2. Do you consider the topic original or relevant in the field? Does it address a specific gap in the field?

Authors should describe contribution and significance of this study.

3. What does it add to the subject area compared with other published material?

Authors should describe contribution and significance of this study.

4. What specific improvements should the authors consider regarding the methodology? What further controls should be considered?

First, this study should describe detailed respondents select strategy. Second, does any bias and limitation in the model? Third, please develop a theoretical framework to be able to analyze the consistency of the outputs with the framework.

5. Are the conclusions consistent with the evidence and arguments presented and do they address the main question posed?

Yes, however this study lacks detailed discussion.

6. Are the references appropriate?

This study needs some new literature.

7. Please include any additional comments on the tables and figures.

Too many redundant tables. Some tables can combine a table, such as Table 8-11.

Author Response

(The authors gave the same response as above.)
